# The Potential of Instrumental Insemination for Sustainable Honeybee Breeding

**DOI:** 10.3390/genes14091799

**Published:** 2023-09-14

**Authors:** Manuel Du, Richard Bernstein, Andreas Hoppe

**Affiliations:** Institute for Bee Research Hohen Neuendorf, Friedrich-Engels-Str. 32, 16540 Hohen Neuendorf, Germany; richard.bernstein@hu-berlin.de (R.B.); andreas.hoppe@hu-berlin.de (A.H.)

**Keywords:** instrumental insemination, honeybee breeding, sustainable breeding, inbreeding, genetic gain

## Abstract

Mating control is crucial in honeybee breeding and commonly guaranteed by bringing virgin queens to isolated mating stations (IMS) for their nuptial flights. However, most breeding programs struggle to provide sufficiently many IMS. Research institutions routinely perform instrumental insemination of honeybees, but its potential to substitute IMS in breeding programs has not been sufficiently studied. We performed stochastic simulations to compare instrumental insemination strategies and mating on IMS in terms of genetic progress and inbreeding development. We focused on the role of paternal generation intervals, which can be shortened to two years with instrumental insemination in comparison to three years when using IMS. After 70 years, instrumental insemination yielded up to 42% higher genetic gain than IMS strategies—particularly with few available mating sites. Inbreeding rates with instrumental insemination and IMS were comparable. When the paternal generation interval in instrumental insemination was stretched to three years, the number of drone producers required for sustainable breeding was reduced substantially. In contrast, when shortening the interval to two years, it yielded the highest generational inbreeding rates (up to 2.28%). Overall, instrumental insemination with drones from a single colony appears as a viable strategy for honeybee breeding and a promising alternative to IMS.

## 1. Introduction

In recent years, breeding of honeybees (*Apis mellifera*) has gained significant popularity in many parts of the world. Typically, honeybees are bred for production traits, such as honey yield; manageability traits, such as gentleness; and health traits, such as resistance against the parasite *Varroa destructor* [1,2,3,4]. Several breeding endeavors concern particularly small local populations of regionally adapted honeybees, which are in danger of replacement or hybridization with other subspecies [5,6,7]. The goal is to increase their attractiveness to local beekeepers by improving economically relevant features. It is summarized by the expression ‘*conservation by utilization*’ [8,9]. In order to avoid genetic crossings with other subspecies, it is mandatory to control the mating process of honeybee queens. Hereby, also mating with undesired drones of the own subspecies is prevented, which is crucial for genetic progress in honeybee breeding programs [10,11,12,13,14]. Newly hatched queens typically perform one or several nuptial flights on which they mate with drones from nearby colonies. To gain control over the drones that are involved in the mating process, one can establish isolated mating stations (IMS, a survey of all abbreviations and variables can be found at the end of this article). There, a sister group of drone producing queens (DPQ) is located at a geographically remote area that is otherwise void of honeybee colonies [10,15,16]. Virgin queens are brought to these IMS for their nuptial flights and consequently only mate with drones that share a common grand-dam. This grand-dam (sometimes called *4a-queen* [17,18,19]) is usually selected for her superior genetics. In this way, a selection process on the paternal path is ensured and valuable pedigree information is gained to perform genetic evaluations [20,21]. However, the implementation and maintenance of IMS are costly, and suitable geographic localities are often scarce, so only few mating stations can be entertained for a breeding population [15,22,23]. This turns IMS into a bottleneck for effective population sizes and inbreeding control. As shown by a recent simulation study [24], a breeding population of 200 honeybee colonies per year would need at least 12 IMS for a sustainable selection process. Populations with 1000 colonies per year should ideally be provided with no less than 40 IMS. In practice, these requirements can rarely be met, hence alternative modes of mating control need to be sought.

In many agricultural species, instrumental insemination has become the default reproduction technique as it provides maximum control over the mating process. The first believable reports of successful instrumental inseminations of honeybee queens date back to the early 20th century [25]. In the meantime, the procedures have been developed to work consistently and standard methods have been established [26]. However, still only a minority of breeders rely on this mode of mating control. In European honeybee breeding programs registered on www.beebreed.eu, accessed on 8 September 2023, approximately 17% of queens are instrumentally inseminated, while the use of IMS remains the predominant mating strategy [4]. However, breeders’ preferences are distributed heterogeneously, and the example of Poland, where reportedly more than 90% of honeybee breeders rely on instrumental insemination [27], shows that it is possible to create breeding programs based entirely on this strategy.

From a theoretical point of view, instrumental insemination of honeybees is expected to have several potential advantages in comparison to mating on IMS. Even by carefully selecting suitable areas for IMS, the possibility of foreign drones participating in a mating can never be ruled out completely [15,22]. Thus, instrumental insemination provides a greater security regarding the genetic quality of drone material and correctness of pedigrees [10]. Furthermore, to provide the necessary density of drones on an IMS, several DPQ are required for the mating process [28,29,30], and the production, transport, and colony care to maintain a high number of drones are challenging and costly [31,32]. Once an IMS is established and in use, for queens mating there, it can neither be detected which individual colony a mating drone came from nor how many drones the queen mated with. Consequently, only probabilistic methods of relationship calculations are possible based on the fact that the DPQ on an IMS share a common dam [20,21,33]. In instrumental inseminations, the drones for one queen can be taken directly from only one colony (Figure 1) [34,35]. Hereby, the genetic superiority of mating drones is guaranteed more reliably, and, in addition, relationships within the population can be estimated with greater precision [36]. Both aspects are likely to improve genetic response. Moreover, implementation of instrumental insemination with drones from a single colony can also have a positive influence on generation intervals. On IMS, the paternal generation interval typically amounts to three years [17]: Honeybee colonies are usually performance tested in the year after the queen hatches so that they can be selected based on estimated breeding values after two years. If a queen is selected to produce the DPQ of an IMS, it generally takes another year until the drone producing colonies will supply the necessary number of drones. If drones for instrumental inseminations are taken directly from selected colonies, the paternal generation interval can be reduced to two years, which is also the standard age difference between a queen and her dam (maternal generation interval) [17,37]. Evidently, shorter generation intervals are concomitant with faster genetic progress.

It should be mentioned, however, that only a fraction of breeders who practice instrumental insemination actually exhaust these potential advantages. Following the observation that colonies with more diverse paternal ancestry often display greater vitality and health [38,39,40], many breeders decide against inseminating queens with drones from only one colony and instead emulate the situation on an IMS. By doing so, of the potential benefits of instrumental insemination, only the secure exclusion of foreign drones prevails. Some breeders also refrain from single colony inseminations because they apprehend high inbreeding rates. Actually, the influence of instrumental insemination with drones from a single colony on inbreeding development under truncation selection is ambiguous. On the one hand, the more reliable selection and closer relationships between sister queens may yield higher inbreeding rates; on the other hand, by inseminating different queens with drones from many genetically different colonies, a greater genetic variety and thus lower inbreeding can be maintained in the population. On www.beebreed.eu, approximately a fourth of all instrumentally inseminated queens are fertilized with drones from a single colony.

Computer simulation studies have been used to search for optimized breeding schemes in honeybees since the 1980s [41,42]. In recent years, they focused on the relevance of controlled mating [11,12], the role of assumed genetic parameters [43], the inclusion of genomic information [44,45,46], the influence of the queens’ polyandry [35], or the general trade-off between genetic progress and inbreeding [24,47]. However, with the exception of the study by Kistler et al. [35], all recent theoretical studies on honeybee breeding rely on IMS as means of mating control. It is therefore unknown how the theoretical advantages of instrumental insemination with drones from a single colony over IMS quantify in various breeding scenarios. In the following, we present simulations on honeybee breeding based on both mating strategies. We investigate how the choice of mating strategy influences genetic progress and inbreeding development under different population sizes and genetic parameters. Hereby, we place a specific focus on the role of potentially shortened paternal generation intervals under instrumental insemination.

## 2. Materials and Methods

### 2.1. Genetic Model

Breeding populations of honeybees were simulated with the program BeeSim [47]. Honeybee populations of two different sizes ($N_{Q}=500$ and $N_{Q}=1000$ breeding queens per year, not counting DPQ on IMS) were simulated over the course of 70 years. These population sizes fall well into the range of breeding populations registered in www.beebreed.eu, which span from 120 (*A. m. ruttneri*, Malta) to over 8000 queens per year (*A. m. carnica*, main population). Queens were selected for a purely additive trait. Since honeybee traits are typically influenced by both queen and workers [1,2,3,4], the trait was modeled with (maternal) queen effects and (direct) worker effects. Genetic parameter estimates for various traits in different honeybee populations suggest that direct variances typically exceed maternal variances and that maternal and direct effects are negatively correlated [2,4]. Following these observations, we considered two different sets of genetic parameters for the selection trait. In the base population, both traits followed maternal and direct genetic variances of $\sigma_{A,m}^{2}=1$ and $\sigma_{A,d}^{2}=2$, respectively, while the residual variance for both traits was set to $\sigma_{E}^{2}=4$. They only differed in the correlation $r_{md}$ between maternal and direct effects, where either a low ($r_{md}=-0.18$) or medium ($r_{md}=-0.53$) negative correlation was assumed.

According to [47], long-term simulation studies in honeybee breeding should use finite locus genetic models rather than Fisher’s infinitesimal model, because the latter does not account for the loss of favorable alleles by genetic drift [48]. Traits were thus chosen to be genetically determined by 400 unlinked biallelic loci. Regarding the allele frequencies, standard quantitative genetic theory [49,50] and observations in real farm animal data [51] point at U-shaped distributions. Accordingly, we let the initial allele frequencies follow a (U-shaped) β(0.5,0.5)-distribution. The distribution of allele effects is often assumed to be L-shaped and heavy-tailed [52,53]. Following [24,47,54], we formed it as a weighted mixture of multivariate Laplace and normal distributions. Since loci were unlinked, an ad hoc formation of the base population was possible. Later generations inherited their genetic setups based on the Mendelian rules.

Every simulated year, we updated two separate relationship matrices. True relationships between queens were calculated following the algorithm of Fernando and Grossman [55] for X-chromosomal inheritance, which is often applied in honeybee contexts [33,35,56]. These relationships were used to report inbreeding coefficients of queens. However, true relationships after Fernando and Grossman require knowledge about the individual drone sires of queens and are not available in real life. To facilitate realistic genetic evaluations, we thus maintained a second matrix of accessible relationships. Here, relationships were calculated based on recorded honeybee pedigrees following the probabilistic methods of [21,33]. The inverse of the accessible relationship matrix was used as input data for genetic evaluations.

### 2.2. Selection

At the age of one year, a performance test was simulated for queens and their colonies and, subsequently, a best linear unbiased prediction (BLUP) breeding value estimation was performed with the BLUPF90 program (version 1.63) [57]. For breeding value estimations, colonies were randomly assigned to one of 40 ($N_{Q}=500$) or 80 ($N_{Q}=1000$) apiaries, each of which was treated as a fixed effect. To account for genetic drift, the genetic variances used for the BLUP process were reassessed from the population every five years [24]. Based on their estimated breeding values, the best 20% of two-year-old queens were selected to each produce five queens of the next generation. As it is standard for queen production, the selection criterion for a queen was formed as the sum of maternal and direct breeding values of her worker group, thereby constituting the expected genetic quality of an offspring queen [20,58,59].

### 2.3. Controlled Mating

Unlike IMS, instrumental insemination is not fixed to a specific location and can therefore be organized in various ways. Breeders may visit an inseminator or *vice versa*, insemination events may be held, or experienced breeders can inseminate their queens by themselves. However, for better comparison of the fertilization strategies, we modeled all instrumental inseminations to take place at fixed instrumental insemination stations (IIS). While an IIS comprised multiple drone producing colonies, the drones for an individual insemination were always taken from only one colony (single colony inseminations). We use the term *fertilization station* to mean either an IMS or an IIS. Four types of fertilization stations were implemented in the simulations. These were:IMS,IIS with a two-year paternal generation interval (called IIS_2_),IIS with a three-year paternal generation interval (called IIS_3_),IIS with a mixed paternal generation interval (called IIS_mix_).

All queens of a population were mated at fertilization stations of the same type; no mixed strategies were implemented. For brevity, we may thus use the names of the different fertilization stations synonymous with the associated breeding strategies (i.e., ‘*strategy IMS*’ for ‘*the breeding strategy relying on IMS for mating control*’, and so on). Three different numbers $N_{\mathrm{FertS}}$ of fertilization stations per year were considered:$N_{\mathrm{FertS}}=5$, representing a realistic number for small-scale honeybee breeding programs (e.g., *A. m. mellifera* in Switzerland [60]),$N_{\mathrm{FertS}}=20$, which was found to be a theoretical minimum number for successful and sustainable breeding with IMS [24],$N_{\mathrm{FertS}}=50$, representing the ideal situation of an abundance of fertilization stations.

All queens with the same dam were mated at the same fertilization station.

In simulations with IMS, the $N_{\mathrm{FertS}}$ best three-year-old queens were selected to produce a sister group of eight DPQ, the setup of one IMS. Queens that mated on an IMS were paired with 12 drones, whose dams were chosen randomly among the DPQ. While 12 drones mating a queen is reported as an average value in the literature [61], eight DPQ on a mating station is generally seen as a minimum requirement for sufficient drone production [28,29]. The simulated mating and selection procedures were identical to those used in earlier simulation studies on honeybee breeding [24,47,59].

In IMS-based breeding strategies, two factors influence the selection intensity on the paternal path: the number $N_{\mathrm{FertS}}$ of mating stations and the number $N_{\mathrm{DPQ}}$ of DPQ per mating station ($N_{\mathrm{DPQ}}=8$ in our simulations). In contrast, when IIS are employed and queens are inseminated with drones from one colony, only the total number of queens used as drone producers is relevant, and it is secondary if there are few IIS with many DPQ or many IIS with few DPQ. We chose IIS to be composed of eight drone producers each. In practical implementation, this means that the effort to maintain an IIS as specified here is roughly equal to the effort to emulate an IMS via instrumental insemination.

Thus, when matings were performed on IIS, each year $8\cdot N_{\mathrm{FertS}}$ queens were selected as drone producers and assigned randomly to the IIS so that each IIS comprised eight queens. Here, the selection criterion was defined as the sum of maternal and direct estimated breeding values of the queen, because the genetic quality of (haploid) drone offspring is determined by the queen alone [18]. Each virgin queen was inseminated with 12 drones that all came from the same randomly chosen queen on the IIS. However, to counteract high inbreeding rates, it was prevented to inseminate queens with drones from their own dam or an aunt. The three different types of IIS (IIS_2_, IIS_3_, and IIS_mix_) only differed in the group of queens from which the drone producers were selected. These were all two-year-old queens, all three-year-old queens, and all two-or-three-year-old-queens, respectively (Figure 2).

No queen losses were simulated, so all queens from the relevant age cohorts were actually available for selection. While this may not be a realistic setup (particularly for strategy IIS_3_), simulating the different types of IIS in this way allowed us to effectively separate the effects of shortened generation interval and higher precision in the breeding value estimation when comparing different mating strategies. The combinations of 2 population sizes, 2 sets of trait parameters, 4 types of fertilization stations, and 3 different numbers of fertilization stations constituted a total of 48 simulation setups. Simulations for each of these setups were repeated 200 times to achieve stable results.

### 2.4. Data Analysis

The statistic analysis of the simulation output was performed with the software R (version 4.2.2) [62] using the *tidyverse* family of packages [63]. For each repetition of the simulation and each simulated year, true breeding values and inbreeding coefficients were averaged over the $N_{Q}$ colonies of that year. Hereby, the true breeding value of a colony was taken to be the sum of the queen’s maternal and the worker group’s direct breeding value (so-called *performance criterion* [59]), and the inbreeding coefficient was that of the queen. In setting IIS_mix_, we further calculated the average ages of the selected drone producers (i.e., paternal generation intervals). Subsequently, we calculated means and standard deviations of these averaged values over the 200 repetitions for each setting. While the mean values report the expected behavior of breeding strategies and thus constituted the center of our interest, the standard deviations allowed us to perform statistical tests to underline the significance of observed differences between strategies. Statistical tests were carried out in the form of Welch’s *t*-test [64].

### 2.5. Effective Number of Sires

In animal breeding, important statistics of a selection program, such as the effective population size or the selection intensity, can be estimated by the numbers of selected dams and sires per generation. For example, Wright’s [49] classic formula for the effective population size $N_{e}$ reads
$$N_{e}=\frac{4N_{d}N_{s}}{N_{d}+N_{s}},$$
where $N_{d}$ and $N_{s}$ are the generational numbers of selected dams and sires, respectively. In a honeybee breeding context, particularly when comparing different methods of mating control, the question arises, what should be counted as a *sire*? To answer this question, we derive the concept of an *effective number of sires* for honeybees.

Traditionally, the group of DPQ on an IMS is seen collectively as a *pseudo-sire* [20,47,65]. However, this notion stems mainly from the role of this entity in pedigrees, not necessarily from its influence on effective population sizes or related parameters. After all, a pseudo-sire is made up of genetically different individual queens. On the other hand, counting the DPQ on an IMS as individual sires would neglect the genetic similarity between them, as they are all sisters. With instrumental insemination, the notion appears more straightforward. Because drone producers on an IIS are not structurally related, they are seen as individual sires. This coincides with the role they take in a honeybee pedigree [18].

To arrive at a meaningful notion of effective numbers of sires, we assume a group of $N_{s}$ unrelated and non-inbred queens that (via their drones) each sire a large number of queen offspring. If we randomly choose two offspring queens and draw their paternally inherited alleles at an arbitrary fixed locus, the probability of these two alleles to be identical by descent (ibd) is
$$p_{\mathrm{pat},\mathrm{ibd}}=\frac{1}{2N_{s}}.$$

The reason is the following: Since the $N_{s}$ sires are assumed unrelated, two offspring can only inherit ibd alleles if they share the same sire (probability: $\frac{1}{N_{s}}$). Furthermore, because the sires are non-inbred, ibd alleles in the offspring further require that the same allele was passed on to both (probability: $\frac{1}{2}$).

For a general group of sires, we can determine the probability $p_{\mathrm{pat},\mathrm{ibd}}$ for two of their offspring inheriting ibd alleles and then define their effective number as
$$N_{s,\mathrm{eff}}=\frac{1}{2p_{\mathrm{pat},\mathrm{ibd}}}.$$

For $N_{\mathrm{FertS}}$ IIS with $N_{\mathrm{DPQ}}$ drone producers each, it is conceivable that all $N_{\mathrm{FertS}}\cdot N_{\mathrm{DPQ}}$ drone producers are non-inbred and mutually unrelated. Then, their effective number coincides with the total number: $N_{s,\mathrm{eff}}=N_{\mathrm{FertS}}\cdot N_{\mathrm{DPQ}}$.

We turn to the case of DPQ on IMS. Again, assume that there are $N_{\mathrm{FertS}}$ IMS, each comprising $N_{\mathrm{DPQ}}$ DPQ. Ideally, all DPQ are non-inbred, and DPQ of different IMS are unrelated. The average relationship of DPQ on the same mating station is denoted by $a_{ss}$ and typically assumed in the vicinity of 0.4 [3,4,58,65,66]. If we take two offspring that were sired on these IMS, the value $p_{\mathrm{pat},\mathrm{ibd}}$ calculates as follows: First, for two paternally inherited alleles to be ibd, they have to come from the same IMS, because the DPQ of different IMS are unrelated (probability: $\frac{1}{N_{\mathrm{FertS}}}$). If that is the case, they can either come from the same DPQ (probability: $\frac{1}{N_{\mathrm{DPQ}}}$) or from different DPQ (probability: $1-\frac{1}{N_{\mathrm{DPQ}}}$). In the former case, the probability that the same allele was inherited is $\frac{1}{2}$, while in the latter case, the probability of the two drawn alleles to be equal is $\frac{a_{ss}}{2}$ by the definition of the relationship coefficient $a_{ss}$ [67] (p. 178). In total, this leaves us at
$$p_{\mathrm{pat},\mathrm{ibd}}=\frac{1+a_{ss}\left(N_{\mathrm{DPQ}}-1\right)}{2N_{\mathrm{FertS}}\cdot N_{\mathrm{DPQ}}},$$
and consequently
$$N_{s,\mathrm{eff}}=\frac{N_{\mathrm{FertS}}\cdot N_{\mathrm{DPQ}}}{1+a_{ss}\left(N_{\mathrm{DPQ}}-1\right)}.$$

In our simulations with $N_{\mathrm{DPQ}}=8$ and $a_{ss}\approx 0.4$, the effective number of sires in IIS strategies was thus $8\cdot N_{\mathrm{FertS}}$, while in IMS strategies it was approximately $2.1\cdot N_{\mathrm{FertS}}$. In other words, an IMS as we simulated it effectively counts as slightly more than two sires.

## 3. Results

### 3.1. Genetic Progress

The genetic gain after 70 years ranged from 10.13 units ($N_{Q}=500$, $r_{md}=-0.53$, 50 IIS_3_) to 22.85 units ($N_{Q}=1000$, $r_{md}=-0.18$, 5 IIS_mix_). *Ceteris paribus*, the genetic gain in the larger population ($N_{Q}=1000$) was between 5.6% and 19.9% higher than in the smaller population ($N_{Q}=500$); likewise, the trait with $r_{md}=-0.18$ had a 40.0% to 50.0% higher revenue than the trait with $r_{md}=-0.53$ (Figure 3).

For all traits, population sizes, and selection intensities, strategy IIS_mix_ yielded the highest genetic gain, significantly ($p<10^{-19}$) outperforming the next-best strategy by 4.6% to 13.2% after 70 years. The ranking of the other strategies depended on $N_{\mathrm{FertS}}$. When only 5 fertilization stations were simulated, both IIS_2_ and IIS_3_ led to similar genetic gain after 70 years, between 14.6% and 25.9% higher than what was achieved with IMS. Comparing IIS_2_ and IIS_3_, we observed that in early years, the shortened generation interval of IIS_2_ yielded faster genetic progress (11.6% to 13.1% higher after 20 years) but slowed down considerably over time so that after 70 years both strategies differed by less than 0.35 units. Throughout, genetic response with strategy IIS_2_ showed the behavior of a step function with genetic progress being achieved only in every second year. All of the above observations showed consistently in both traits and for both population sizes $N_{Q}$.

Increasing the number of fertilization stations to $N_{\mathrm{FertS}}=20$ or $N_{\mathrm{FertS}}=50$ significantly reduced the genetic progress in all IIS-based strategies (p<0.011). In contrast, when using IMS, we observed that the genetic gain was only mildly affected by the number of fertilization stations and reached a maximum for $N_{\mathrm{FertS}}=20$. In consequence, the gap between insemination strategies and IMS became smaller for greater $N_{\mathrm{FertS}}$ and for $N_{\mathrm{FertS}}=50$, strategy IMS even proved significantly superior to IIS_3_ ($p<10^{-12}$).

### 3.2. Inbreeding Coefficients

In all settings, noticeable inbreeding accumulated over time (Figure 4). After 70 years, average inbreeding coefficients ranged from 0.046 ($N_{Q}=1000$, $r_{md}=-0.18$, 50 IIS_3_) to 0.549 ($N_{Q}=500$, $r_{md}=-0.53$, 5 IIS_2_). With all other parameters equal, inbreeding in the smaller population ($N_{Q}=500$) was 2.4% to 18.9% higher than in the larger population ($N_{Q}=1000$). Similarly, the trait with medium negative correlation between maternal and direct effects ($r_{md}=-0.53$) yielded 6.6% to 21.3% higher inbreeding coefficients than the trait with weak negative correlation ($r_{md}=-0.18$). The higher the number of fertilization stations $N_{\mathrm{FertS}}$, the lower the inbreeding rates: after 70 years, average inbreeding coefficients ranged from 0.189 to 0.549 for $N_{\mathrm{FertS}}=5$, from 0.085 to 0.291 for $N_{\mathrm{FertS}}=20$, and from 0.046 to 0.161 for $N_{\mathrm{FertS}}=50$.

Comparing the different types of fertilization stations revealed that with fixed $N_{Q}$, $r_{md}$, and $N_{\mathrm{FertS}}$, strategy IIS_2_ always led to the highest inbreeding coefficients, while strategy IIS_3_ yielded the lowest inbreeding. After 70 years, the average inbreeding in strategy IIS_2_ was 2.4-fold to 2.8-fold higher than in strategy IIS_3_. Comparing the remaining strategies, IIS_mix_ and IMS, showed a dependence on the number of fertilization stations. For $N_{\mathrm{FertS}}=5$, IIS_mix_ yielded clearly higher inbreeding (by 37.5% to 47.9%); this gap narrowed for $N_{\mathrm{FertS}}=20$ (2.6% to 8.4%, still significant with p<0.025). For $N_{\mathrm{FertS}}=50$, the inbreeding with IIS_mix_ was 14.6% to 15.1% lower than with strategy IMS. In all settings, the gap between IIS_3_ and the two other IIS-based strategies in terms of inbreeding was substantial.

### 3.3. Generation Intervals and Inbreeding Rates

An important parameter of a breeding program is the increase of inbreeding per generation [24,68]. To assess this value, it was necessary to determine generation intervals in the respective breeding strategies. Independent of the mode of mating control, the maternal generation interval was always two years. The strategies IMS, IIS_2_, and IIS_3_ had clearly defined paternal generation intervals of three, two, and three years, respectively. The averaged generation intervals were thus 2.5 years for strategies IMS and IIS_3_, and 2 years for strategy IIS_2_. In strategy IIS_mix_, paternal generation intervals were shaped by the respective selection frequencies of two- and three-year-old queens as drone producers.

In early years, the average paternal generation interval in strategy ISS_mix_ oscillated strongly from year to year, taking on values between 2.00 and 2.55 (Figure 5). Later, it stabilized around average values between 2.23 ($N_{Q}=1000$, $r_{md}=-0.18$, $N_{\mathrm{FertS}}=5$) and 2.36 ($N_{Q}=500$, $r_{md}=-0.53$, $N_{\mathrm{FertS}}=50$).

For a population with average generation interval GI and average inbreeding coefficient after 70 years $F_{70}$, we then calculated inbreeding rates according to the formula [24].
$$\Delta F=1-{\left(1-F_{70}\right)}^{\frac{GI}{69}}.$$

The results are given in Table 1. They indicate that for fixed $N_{Q}$, $r_{md}$, and $N_{\mathrm{FertS}}$, the three strategies IMS, IIS_2_, and IIS_mix_ yielded similar inbreeding rates, while the inbreeding rates for strategy IIS_3_ were substantially lower. With IIS_3_, all inbreeding rates were below 1% per generation, whereas the other strategies led to inbreeding rates greater than 1% in the case of $N_{\mathrm{FertS}}=5$.

## 4. Discussion

### 4.1. Explanation of Observations

#### 4.1.1. Genetic Progress

The general observation that larger populations and weaker negative genetic correlation between maternal and direct effects lead to higher genetic gain are standard findings in animal breeding and have also been observed in previous simulation studies on honeybees [24,35,47]. The increased genetic progress for IIS strategies compared to IMS schemes, particularly for $N_{\mathrm{FertS}}=5$, had multiple reasons. First, it should be noted that for $N_{\mathrm{FertS}}=5$, the effective number of sires in IIS-based breeding was 40, as opposed to 10.5 in IMS-based strategies. In this sense, IMS schemes incorporated the sharper selection regiment. However, the sires in IIS strategies were selected with a higher accuracy due to the more precise calculation of the numerator relationship matrix when all mating drones came from the same colony [21,33]. Furthermore, the direct selection of drone producers from the breeding population served as a form of risk reduction. If a queen was wrongfully selected for this purpose (because she had a high estimated breeding value but low true breeding value), the reproduction of her drones only affected the offspring of queens mated with drones from her colony. If, in contrast, the dam of the DPQ of an IMS was selected despite a low true breeding value, this had a negative impact on the offspring of all queens that mated on this mating station.

IIS_2_ initially outperformed IIS_3_ in terms of genetic progress because of the shorter paternal generation interval. The flattening in later years is to be explained by the depletion of genetic variance due to high inbreeding rates, an effect also observed in previous simulation studies [24,47].

When 20 or even 50 IIS were equipped with drone producers, the genetic response decreased because of the reduced selection intensity. Note that in a population with 500 queens per year, allowing for 50 IIS with 8 drone producing colonies each means that 80% of all queens (400 out of 500) are selected for drone production.

In strategy IIS_2_, both maternal and paternal generation intervals were two years. Consequently, there was no genetic exchange between queens of even and odd years. In effect, two entirely separate populations were maintained in parallel, which explains that genetic progress was only made every other year.

#### 4.1.2. Inbreeding

As in the case of genetic gain, the general influences of population size and the correlation $r_{md}$ on inbreeding development (higher inbreeding with smaller $N_{Q}$ and stronger negative $r_{md}$) are standard observations that could previously be seen in several other simulation studies [24,43,47].

The high inbreeding rates in strategy IIS_2_ were partly caused by the fact that queens of even and odd birth years were genetically strictly separated, which halved the effective population size. However, also strategy IIS_mix_ led to high inbreeding rates, even though this strategy allowed for genetic admixture between even and odd years. As we had anticipated the risk of high inbreeding rates, we had avoided to inseminate queens with drones from their own dam or an aunt. However, this simple ad hoc strategy to avert high inbreeding coefficients could not prevent overall inbreeding rates being high. The higher inbreeding rates of strategies IIS_2_ and IIS_mix_ in comparison to strategy IMS, despite a higher effective number of sires, indicate that not only the number of selected animals but also the choice of the concrete individuals had a major influence on inbreeding rates. This is in contrast to what has previously been claimed by Uzunov et al. [17].

#### 4.1.3. Paternal Generation Intervals

In strategy IIS_mix_, the average paternal generation interval indicated that between a fourth and a third of the selected sire queens were three years old. This shows that exceptionally good queens may be used for reproduction over an extended period of time, as they remain competitive. The oscillations in the initial years can be explained by the design of the simulations. The queens of the first two years formed a base population and were randomly mated because no drone producers were available yet. Queens of year three were the first who could be inseminated, but only with drones from queens of the first year, so the generation interval necessarily was two years. Queens of year four were inseminated with drones from queens of the two base years. As there are no structural genetic differences between the first 2 years, the expected average paternal generation interval is 2.5 years. Queens of year five were inseminated with drones of queens from years two (base population) or three (first generation that benefited from selection). Evidently, there was a strong favor for the latter. These considerations may be continued, but the differences in selected generations between years will become less and less pronounced, thus evening out the oscillations.

### 4.2. Implications for Real Honeybee Breeding Programs

#### 4.2.1. Sustainability

As for all agricultural species, selective breeding in honeybees leads to accumulation of inbreeding and reduction of genetic diversity. A lower genetic variance means less potential for natural selection and adaptation to changing environments and other stresses [69]. For this reason, some authors disapprove of selective honeybee breeding altogether [70]. In livestock, it is consensus that a certain rate of inbreeding is acceptable because its effects are alleviated by mutation and recombination [71]. Based on rough estimates and calculations [72,73], the Food and Agriculture Organization of the United Nations (FAO) defines breeding systems as sustainable if the generational increase of inbreeding is below 1% [68]. This threshold is applied for a wide variety of domesticated species [74,75,76]. In haplodiploid organisms, the consequences of inbreeding differ from diploid species. The genome of honeybees, like many other haplodiploids [77], contains a sex determining locus which must be heterozygous for an egg to develop into a worker bee [78]. Since high inbreeding increases the probability for homozygosity at this locus, it may lead to scattered brood and thus weakened colonies [79]. On the other hand, inbreeding effects may be reduced because deleterious recessive alleles are expressed in the haploid drones and therefore get purged [80]. It is thus unclear, if the same threshold as for diploid species should be applied to haplodiploids [81] (p. 140). However, due to the lack of a better notion of sustainability, the 1% definition of the FAO has previously been applied to honeybee breeding schemes [24].

Following this definition, breeding with IMS was sustainable when there were at least 20 mating stations. This is in line with earlier findings of Plate et al. [24]. With 20 IIS à eight DPQ, also all instrumental insemination strategies yielded sustainable results. The breeding scheme IIS_3_ showed that in theory it is possible to build sustainable breeding programs with instrumental insemination and only few IIS. But in practice, it is questionable if enough three-year-old queens will still be alive and thus available for drone production. Only in 22% of single colony inseminations currently registered in www.beebreed.eu, the drone producing queen was aged three years or older. However, we expect that by use of within-family selection [35,42] or Optimum Contribution Selection [82], sustainable inbreeding rates could be achieved without relying heavily on old drone producers to be available.

#### 4.2.2. IIS as an Alternative to IMS

By its nature, a simulation study will always be a simplification of reality. We therefore could not factor in all influential aspects of real-life honeybee breeding. In existing breeding programs such as those managed under www.beebreed.eu, IMS and IIS are used in parallel and the selection intensities differ widely. Breeders tend to place their focus on different selection traits and from time to time introduce new stock into the population [2,4]. These practices work towards a reduction of inbreeding rates. On the other hand, some breeders use their best queens as dams extensively, thus creating heterogeneous sister group sizes and higher risks of inbreeding. The extent to which such further aspects influence the results can only be subject to speculation.

Nevertheless, it is possible to draw real-life conclusions from our findings because they show the extreme ends of possible scenarios. Unlike IMS, instrumental insemination does not rely on geographic features, and IIS can harbor almost arbitrary numbers of sires. Using instrumental insemination, it is thus possible to actually implement breeding schemes that are theoretically found to be optimal. In this regard, our simulations clearly indicate that instrumental insemination can serve as a substitute for IMS and that it is not necessary to emulate the situation on an IMS; instead, drones may be taken from a single colony. This practice will lead to higher genetic progress at comparable inbreeding rates.

It should further be noted that strict truncation selection for estimated breeding values, as we simulated it, is rarely practiced. Particularly for small and endangered honeybee populations, breeders will have a strong focus on subspecies preservation and thus incorporate strategies of inbreeding avoidance in their decisions. For example, www.beebreed.eu, accessed on 8 September 2023, allows to check in advance if a certain mating will lead to high inbreeding coefficients, a tool that is regularly used by breeders [17]. By use of instrumental insemination with drones from a single colony, these strategies can become more efficient because of the increased accuracy of relationship calculations. Strategy IIS_3_ further hints at the possibility to drastically decrease inbreeding rates if many three-year-old queens are available for selection as drone producers. However, in most real-world breeding systems, this is currently not the case.

#### 4.2.3. Diversity of Drones

Several studies have shown that colonies with genetically diverse worker bees have fitness advantages in terms of colony strength [38], disease resistance [83], foraging ability [84], and general survival rates [39]. For the traditional breeding trait *honey yield*, Neumann and Moritz [85] found a (non-significant) positive correlation with the queen’s level of polyandry. Therefore, it may seem surprising that we focused on insemination with drones from a single colony, because closely related drones will transmit less genetic variance to the worker group.

However, the ultimate goal of breeding is not that the current generation of queens has strong colonies, but rather that the next generation of queens has good genetic properties. This can be seen by the fact that the selection criterion for a queen is the predicted genetic quality of her daughters [20,58,59]. However, a daughter queen has only one father drone, and her genetic quality is not affected by the other drones her dam mated with. In other words, the fitness advantages of colonies due to their queens’ diverse mating partners are not heritable and thus irrelevant for breeding [13]. By inseminating a queen with drones from only one colony, her daughters’ genetic qualities can be predicted most accurately.

#### 4.2.4. Genetic Model

The genetic model for a queen in our simulations consisted of 400 unlinked loci with purely additive allele effects. This is a vast simplification of reality. In general, it is computationally possible to perform stochastic simulation studies with realistic genomes [86], including that of the honeybee [87]. The recently developed simulation software SIMplyBee [56] builds on AlphaSim [88,89] and aims at highly realistic simulations of honeybee breeding processes. It simulates individual worker bees and therefore allows to model complex interactions between workers with influences on selection traits.

However, the current knowledge on how relevant honeybee traits are influenced by genomics or colony interaction is scarce. Few causative genes have been detected by genome-wide association studies [90,91,92], but often, even relatively simple parameters, such as additive genetic variances, can only be estimated with large standard errors [1,3,93]. When using detailed models of genomes or colony structure, one therefore has to make many assumptions with little information and without a guarantee to actually obtain greater resemblance to reality than our simple model provides. Our genetic model has the advantage to be computationally fast so that we could cover multiple simulation settings with many repetitions.

### 4.3. Limitations of the Study

Despite our findings, it may be ill-advised to hastily dismiss IMS altogether and solely rely on instrumental insemination. In this study, we only investigated the theoretical quantitative genetic aspects of the mating strategies. For a holistic judgment, many more factors would have to be considered. These include, but are not restricted to:

#### 4.3.1. Reproduction Genetics

When queens mate freely in drone congregation areas, thousands of drones compete for mating success, which imposes strong natural selection on reproductional fitness of drones [94]. While this competition is limited on IMS, it is entirely absent in the case of instrumental insemination. The effects of this absence of selection pressure are to date unknown but generally anticipated to be negative [95]. Presumably, this is particularly the case when there are many drones with low flight ability and sperm quality [96].

#### 4.3.2. Other Biological Factors

In addition to genetic factors, other biological factors may influence the attractiveness of mating strategies IMS and IIS. For example, success rates of the mating procedure may differ, and the queen’s productivity and longevity may be affected. In the past, different studies yielded inconclusive results in these respects (see [97] for a survey), but today it is widely believed that when performed correctly, instrumentally inseminated queens have little to no disadvantages to naturally mated queens in these regards [26,97]. Note, however, that these findings mostly do not take the genetic diversity of drones into account. Furthermore, they might be biased by the fact that breeders tend to instrumentally inseminate genetically superior queens at higher rates [98].

#### 4.3.3. Economic Factors

Operating costs for IMS and IIS are likely to differ in a site-specific way. In a large economic assessment of honeybee breeding in Europe, Buechler et al. [99] found large regional differences in the costs for queen rearing and mating in Europe, to a large part caused by differing personnel costs.

Another economic factor is the question of whether sufficiently trained personnel is available. Both IMS and IIS need well-trained staff in order to work properly. On IMS, it requires experience to constantly ensure the necessary density of drones [28], while instrumental insemination requires extensive training because otherwise queens can easily become injured [100].

#### 4.3.4. Sociological Factors

An important aspect that is difficult to monetize is social acceptance. In comparison with other forms of animal husbandry, the public eye generally holds beekeeping and honeybee breeding in high esteem. One reason is that beekeeping is widely seen as a relatively unspoiled and close-to-nature form of agriculture [101], a perception that might suffer when shifting increasingly from natural mating forms to instrumental insemination.

## 5. Conclusions

Instrumental insemination with drones from a single colony appeared as a viable alternative to mating on IMS. At comparable generational inbreeding rates, IIS yielded greater genetic progress. With a prolonged paternal generation interval of three years, much fewer IIS were needed for sustainable breeding than when two-year-old queens were selected as drone producers. From a quantitative genetic point of view, nothing speaks against incorporating IIS into breeding schemes. However, when deciding between IMS and IIS as means of controlled mating, other aspects have to be considered that were not covered by our study. These include biological, economic, and sociological factors.

## Figures and Tables

**Figure 1 genes-14-01799-f001:**
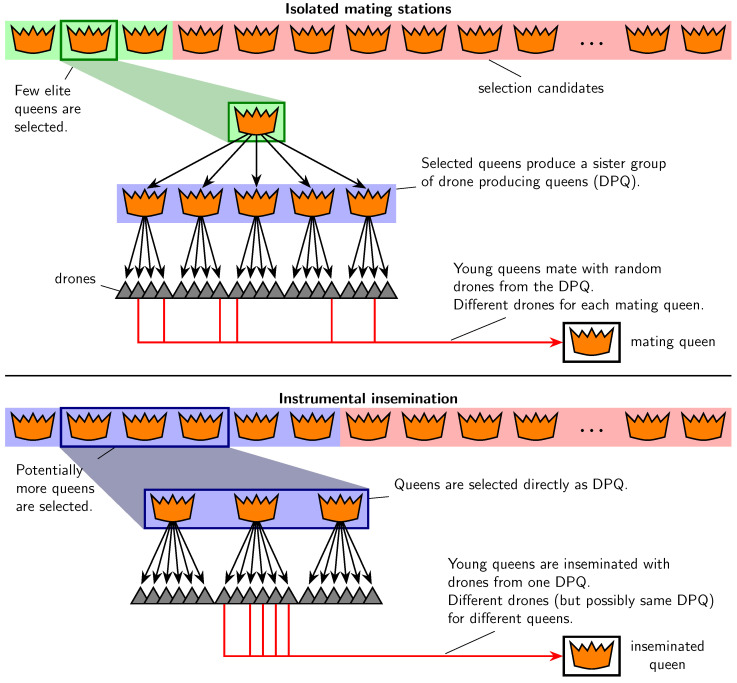
Comparison of the paternal selection paths using isolated mating stations or instrumental insemination with drones from a single colony.

**Figure 2 genes-14-01799-f002:**
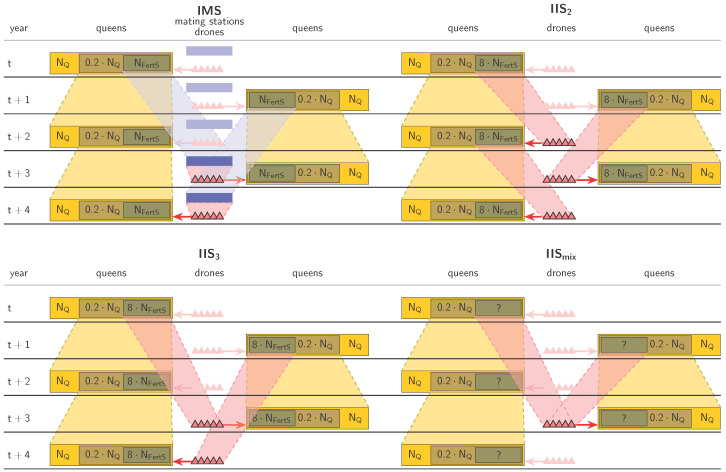
The four different simulated breeding schemes. This figure is inspired by Figure 1 of [47], which was published under a Creative Commons license.

**Figure 3 genes-14-01799-f003:**
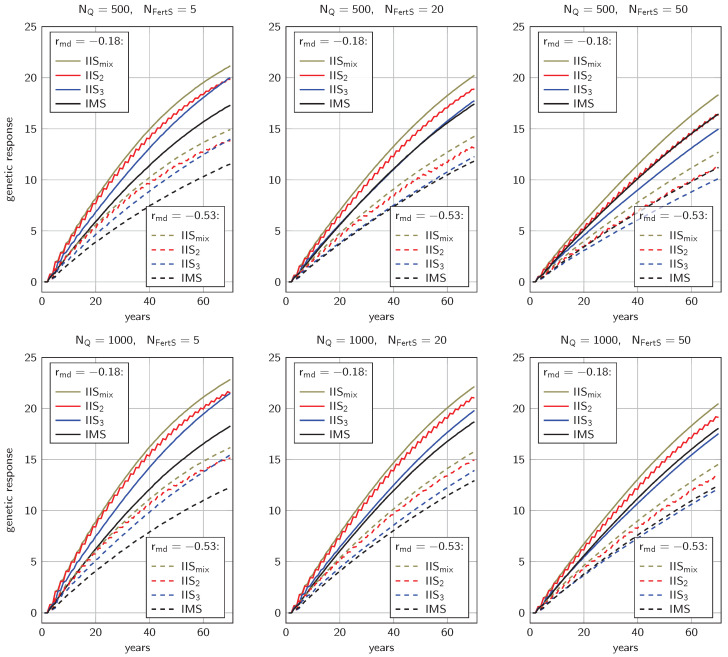
Genetic responses over 70 years of breeding with mating stations or instrumental insemination with drones from a single colony. Results are shown for different population sizes $N_{Q}$ and different numbers of fertilization stations $N_{\mathrm{FertS}}$.

**Figure 4 genes-14-01799-f004:**
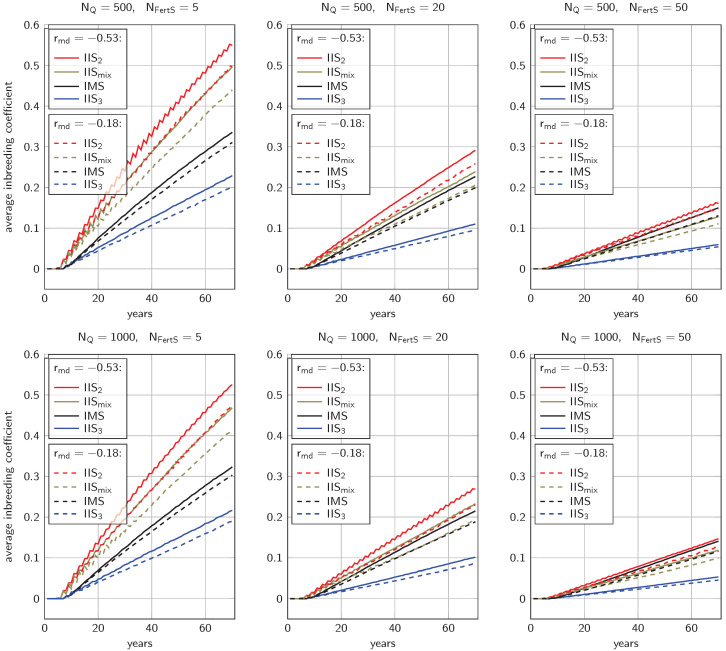
Average inbreeding coefficients over 70 years of breeding with mating stations or instrumental insemination with drones from a single colony. Results are shown for different population sizes $N_{Q}$ and different numbers of fertilization stations $N_{\mathrm{FertS}}$.

**Figure 5 genes-14-01799-f005:**
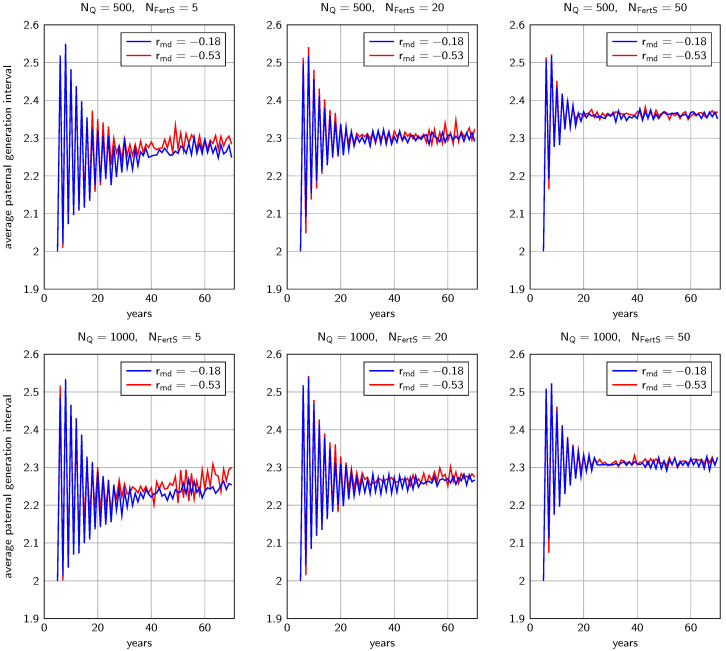
Average paternal generation intervals in strategy IIS_mix_. Results are shown for different population sizes $N_{Q}$ and different numbers of fertilization stations $N_{\mathrm{FertS}}$.

**Table 1 genes-14-01799-t001:** Inbreeding rates in percent. Values exceeding 1% are typeset in bold.

$N_{Q}$	$r_{\mathrm{md}}$	$N_{\mathrm{FertS}}$	IMS	IIS_2_	IIS_3_	IIS_mix_
500	−0.18	5	**1.34**	**1.97**	0.81	**1.77**
500	−0.18	20	0.80	0.86	0.37	0.72
500	−0.18	50	0.51	0.46	0.20	0.37
500	−0.53	5	**1.47**	**2.28**	0.94	**2.10**
500	−0.53	20	0.92	0.99	0.42	0.85
500	−0.53	50	0.59	0.51	0.22	0.43
1000	−0.18	5	**1.30**	**1.86**	0.76	**1.64**
1000	−0.18	20	0.75	0.75	0.32	0.66
1000	−0.18	50	0.45	0.39	0.17	0.33
1000	−0.53	5	**1.40**	**2.14**	0.88	**1.93**
1000	−0.53	20	0.87	0.90	0.39	0.82
1000	−0.53	50	0.55	0.46	0.20	0.39

## Data Availability

An older version of the simulation software BeeSim is available under https://datadryad.org/stash/dataset/doi:10.5061/dryad.1nh544n (accessed on 8 September 2023). Relevant simulation data from this study can be provided by the corresponding author upon reasonable request.

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
