# Peer review of "The Potential of Instrumental Insemination for Sustainable Honeybee Breeding"

_genes, 2023, doi:10.3390/genes14091799_

Round 1

Reviewer 1 Report

The authors present a comprehensive simulation study on honeybee breeding strategies, focusing on genetic progress and inbreeding. It effectively interprets the results, highlighting the impact of population size, correlations, and selection intensities. The discussion successfully links simulation findings to real-world implications. To enhance clarity, subheadings could be used, and a concise concluding paragraph summarizing key findings and practical recommendations would be beneficial. Additionally, minor grammatical and citation-related revisions are needed for improved readability and academic rigor. Overall, the manuscript offers valuable insights into honeybee breeding but would benefit from the suggested enhancements.

Abstract:

the abstract provides a comprehensive overview of the study, it could benefit from posing some specific scientific questions or hypotheses that the research aimed to address. It would be helpful to include a sentence or two in the abstract regarding potential future directions or implications of the research. This could spark interest among researchers and practitioners in the field.

Introduction:

There are several aspects that should be considered for improvement:

the introduction provides a comprehensive overview of the context, it would be beneficial to explicitly state the specific scientific questions or hypotheses that the research aims to address. This would provide readers with a clear understanding of the study's objectives.

Please  rephrase the sentence, "In the European honeybee breeding programs registered in www.beebreed.eu, about 20% of queens are instrumentally inseminated and the use of IMS clearly is the predominant mating strategy," to "In European honeybee breeding programs registered on www.beebreed.eu, approximately 20% of queens are instrumentally inseminated, while the use of IMS remains the predominant mating strategy."

Provide the citations for few more specific references to support some of the claims made. For example, when discussing the advantages of instrumental insemination, you could reference studies or data that support these assertions.

The introduction effectively discusses prior work related to honeybee breeding but could benefit from a brief synthesis of this work, summarizing what has been studied and what gaps in knowledge the current research intends to fill.

The introduction of the manuscript provides a solid foundation for the research by outlining the context and challenges of honeybee breeding. To enhance its impact and readability, consider restructuring complex sentences, explicitly stating scientific questions, making minor grammatical improvements, providing more specific citations, engaging the reader, and synthesizing prior research.

Materials and Methods: 

This section is comprehensive and provides a clear understanding of how the study was conducted. However, there are some points to consider for improvement:

The section is well-structured but contains some complex sentences and technical terms that may be challenging for readers without a deep understanding of honeybee breeding. Breaking down lengthy sentences into smaller, more digestible parts and providing explanations for technical terms could enhance clarity.

It's essential to explicitly state the scientific questions or hypotheses that the research aims to address. This would provide context for readers and help them understand the purpose of the study.

Also, there are some minor grammatical improvements that can be made for fluency and precision. For example, consider rephrasing, "The mating and selection procedure was thus identical to earlier simulation studies on honeybee breeding," to "The mating and selection procedures were identical to those used in earlier simulation studies on honeybee breeding."

It's important to mention any assumptions made during the simulation, such as the assumed values of genetic parameters and relationships between drones, queens, and worker bees. Transparency in assumptions can help readers better interpret the results.

Data Analysis: While the section details the simulation setup, it could benefit from a brief explanation of how the data collected during the simulations were analyzed to derive conclusions. This would provide insight into the statistical methods used.

Justification for Assumptions: Providing brief explanations for certain assumptions made during the simulations, such as the choice of allele frequencies and distribution of allele effects, can help readers appreciate the validity of the model.

Explain why specific parameters, such as the numbers of queens, drones, or fertilization stations, were chosen for the simulations. How do these parameters relate to real-world honeybee breeding scenarios, and what scientific rationale underlies these choices? 

Results:

The results are presented clearly, and the organization of the section is logical, making it easy for the reader to follow the findings. However, it could be beneficial to use subheadings to further segment the information, especially when discussing different aspects of genetic progress and inbreeding.

Statistical Significance: It would be valuable to provide information about the statistical significance of the observed differences in genetic progress, inbreeding coefficients, and generation intervals. Mentioning statistical tests or confidence intervals can strengthen the credibility of the results.

While the section presents numerical results, it would be helpful to discuss any patterns or trends observed in the data. For instance, are there consistent trends across different population sizes, mating strategies, or selection intensities? 

Consider providing more insight into the practical implications of the results. How might the observed genetic progress, inbreeding rates, and generation intervals impact real-world honeybee breeding programs? Are there specific recommendations or considerations for breeders based on these findings?

Consistency in Reporting: Ensure consistency in reporting the results, such as consistently using "NQ" for population size and providing units for genetic progress and inbreeding coefficients where applicable.

While the section is comprehensive, be mindful of conciseness. Ensure that the text provides all necessary information but avoids unnecessary repetition. Additionally, ensure consistency in reporting and consider statistical significance where applicable.

Discussion:

Overall, the discussion is well-structured and provides valuable insights into the study's results. However, I have some comments and suggestions for improvement:

The discussion is logically organized, with clear transitions between different topics. The use of subheadings within the discussion could further enhance the readability and help readers navigate through complex concepts.

Sustainability Threshold: The discussion references the Food and Agriculture Organization's (FAO) definition of sustainable breeding systems based on inbreeding rates, which is a valuable reference point. However, it might be helpful to briefly explain the importance of maintaining inbreeding rates below 1% in honeybee breeding for readers who may not be familiar with this concept.

The discussion appropriately acknowledges that there are additional factors beyond genetic aspects, such as reproduction genetics, biological factors, economic considerations, and sociological factors, that can influence breeding decisions. However, it might be beneficial to expand on the potential impact of these factors on the choice between instrumental insemination and natural mating.

To provide a concise summary of the key findings and their implications, consider adding a brief concluding paragraph to wrap up the discussion.

Reviewer 2 Report

genes-2544191 Reviewer comments

Manuscript genes-2544191: The Potential of Instrumental Insemination for Sustainable Honeybee Breeding

The authors performed stochastic simulations to test three different strategies of instrumental insemination in comparison to mating on isolated mating stations in terms of the quantitative genetic aspects of genetic progress and inbreeding development. After 70 years, instrumental insemination strategies yielded up to 42% higher genetic gain than isolated mating stations strategies – particularly when only few mating sites were available. Inbreeding rates with instrumental insemination and isolated mating stations were comparable. When the paternal generation interval in instrumental insemination was stretched to three years, the number of drone producers required for sustainable breeding schemes was reduced substantially. In contrast, when the generation interval was shortened to two years, it yielded the highest generational inbreeding rates of up to 2.28%. The instrumental insemination with drones from a single colony appears as a viable strategy for honeybee breeding and a promising alternative to isolated mating stations.

The uniqueness of the text is 90% by antiplagiarism.net

The English is almost good but there are misspellings.

The methods and statistics are correct.

There are some mistakes and comments:

1) In the figure 5 - average paternal generaton interval - should be - average paternal generation interval.

2) The authors recommend use drones originated from one colony only. It is known that colonies more resistant and productive with queens mated with many drones. Please discuss following papers - Mattila & Seeley (2007) Genetic Diversity in Honey Bee Colonies Enhances Productivity and Fitness. Science 317:362-364.

Rosenkranz et al., (2010) Biology and control of Varroa destructor. Journal of Invertebrate Pathology 103: S96–S119.

Neumann & Moritz (2000) Testing genetic variance hypothesis for the evolution of polyandry in the honeybee (Apis mellifera). Insectes Sociaux 47:271-279.

3) Discussion part is weak, please add more discussion. Please compare and discuss with recent publications:

Obšteter, J., Strachan, L.K., Bubnič, J. et al. SIMplyBee: an R package to simulate honeybee populations and breeding programs. Genet Sel Evol 55, 31 (2023). https://doi.org/10.1186/s12711-023-00798-y 

Panziera D, Requier F, Chantawannakul P, Pirk CWW, Blacquiere T. The diversity decline in wild and managed honey bee populations urges for an integrated conservation approach. Front Ecol Evol. 2022;10:767950.

Slater GP, Harpur BA. Using genomics to predict drone quality: why are there so many ‘dud’ male honey bees?. In Proceedings of 12th World Congress on Genetics Applied to Livestock Production: 3-8 July 2022; Rotterdam; 2022.

Baumdicker F, Bisschop G, Goldstein D, Gower G, Ragsdale AP, Tsambos G, et al. Efficient ancestry and mutation simulation with msprime 1.0. Genetics. 2022; 220:iyab229.

Faux AM, Gorjanc G, Gaynor RC, Battagin M, Edwards SM, Wilson DL, et al. AlphaSim: Software for breeding program simulation. Plant Genome. 2016;9:3

4) The instrumental insemination can cause more injury to the queens in comparison with mating in isolated mating stations. The instrumental insemination can be successful only if the breeder is a specialist with many years of experience. The breeder in isolated mating stations can be successful without many years of experience. Please note it and discuss.

Please improve the manuscript according to the above comments.

The English is almost well, minor editing is required.
